# Inulin and Chinese Gallotannin Affect Meat Quality and Lipid Metabolism on Hu Sheep

**DOI:** 10.3390/ani13010160

**Published:** 2022-12-31

**Authors:** Zhaohua He, Long Cheng, Shaobin Li, Qiaoling Liu, Xue Liang, Jiang Hu, Jiqing Wang, Xiu Liu, Fangfang Zhao

**Affiliations:** 1Gansu Key Laboratory of Herbivorous Animal Biotechnology, Faculty of Animal Science and Technology, Gansu Agricultural University, Lanzhou 730070, China; 2Faculty of Veterinary and Agricultural Sciences, The University of Melbourne, Dookie College, VIC 3647, Australia

**Keywords:** bioactive compounds, fatty acids, metabolomics, meat quality

## Abstract

**Simple Summary:**

The identification of new and beneficial alternatives to antibiotics is of great value in ruminant farming. The combined addition of inulin and Chinese gallotannin to the diet could reduce the content of saturated fatty acids in mutton while increasing the content of unsaturated fatty acids. Besides, these supplements could also enhance the energy and lipid metabolism of Hu sheep, ultimately improving mutton quality and production performance. Thus, sheep producers are encouraged to consider using these dietary supplements to support better mutton production.

**Abstract:**

The aim of this study was to investigate the impacts of inulin and Chinese gallotannin on the meat fatty acids and urinary metabolites in sheep. Twenty-four healthy (25.80 ± 3.85 kg) weaned Hu lambs of approximately 4.5 months old were equally divided into four groups: control group (basal diet), treatment group I (basal diet + 0.1% inulin), treatment group II (basal diet + 0.1% inulin + 2% Chinese gallotannin), and treatment group III (basal diet + 0.1% inulin + 2% Chinese gallotannin + 4% PEG). The contents of myristic acid (C14:0) and palmitic acid (C16:0) were found to be lower in treatment group II than in the control group (*p* < 0.05). Moreover, the palmitoleic acid (C16:1) content in treatment group II was notably higher than that in the control group (*p* < 0.05), while the elaidic acid (C18:1n9t) content in treatment group II was higher than that in other groups (*p* < 0.05). Besides, the linoleic acid (C18:2n6c) content was higher in the treatment II and control groups than in the treatment I and III groups. Furthermore, compared with the control group, both 4-pyridoxic acid and creatinine in treatment groups I and II were upregulated (*p* < 0.05), while other metabolites, such as nicotinuric acid, l-threonine, palmitic acid, and oleic acid, were drastically downregulated (*p* < 0.05). These differential metabolites were found to be mainly involved in nicotinate and nicotinamide metabolism (ko00760), vitamin B6 metabolism (ko00750), and the fatty acid biosynthesis pathway (ko00061). It is concluded that the combination of inulin and Chinese gallotannin in the diet could improve the energy and lipid metabolism of sheep, which may improve both mutton quality and production performance.

## 1. Introduction

Misuse of feed additives in ruminant production reduces safety and increases environmental pollution in ruminant husbandry. Therefore, there is increased interest in the identification of alternatives to antibiotics, especially in the ruminant production research community [1].

Inulin (C_17_H_11_N_5_), an oligofructose, is widely distributed in over 36,000 plant species such as roots of chicory (15–20%), burdock (3.5–4%), salsify (4–11%), and garlic (9–16%) [2]. Nowadays, most inulin is produced from chicory roots [3,4]. Due to its solubility and capability to alter blood sugar in ruminants, as well as influencing lipid metabolism and mineral absorption, inulin has been approved for use in animal feeds and medications in the USA, Japan, and Europe [5,6]. Recent studies have shown that inulin may improve rumen fermentation, microbial balance, and growth performance in beef cattle [7]. Further, it is considered that inulin can act in combination with other additives (such as inactivated *Bacillus subtilis*, magnesium oxide, and polyphenol-rich pomegranate extracts) and may have additional beneficial effects on the livestock, such as improving immunity [8], regulating sugar and lipid metabolism [9], and reducing endotoxemia [10].

Chinese gallotannin (C_76_H_52_O_46_) is derived from the Chinese gallnut. Chinese gall is an important Chinese herbal medicine, and its main bioactive component is a hydrolyzable tannic acid (TA), gallotannins [11]. It has been reported that TA has biological functions, including anti-inflammatory, antioxidant, antibacterial, and antiviral properties [12,13]. As a hydrolyzed tannin, Chinese gallotannin has significant potential to be used in animal production, due to its high yield from gallnut production [14]. Moreover, low to moderate tannin intake by ruminants can also contribute to reduced CH_4_ emissions, which may enhance the environmental protection goals of the livestock farms [15]. However, excessive tannins intake can have deleterious effects on animals, including reduced food intake and adverse effects on feed digestibility [16]. To minimize the potentially negative effects of tannins, polyethylene glycol (PEG, HO(CH_2_CH_2_O)_n_H) is commonly used as a tannin adsorbent, as it is known to decrease the adverse effects of anorexia in ruminants induced by excessive tannin intake [17,18].

Previous studies have concluded that metabolomics can provide a wealth of information for animal production and health evaluation. Metabolomic analyses can be conducted on a wide range of biological samples, including blood [19], muscle [20], urine [21], feces [22], and intestinal contents [23]. Since inulin is known to interact beneficially with other feed additives for livestock, we speculated that there may be an interactive and beneficial effect between inulin and Chinese gallotannin on animal health and production as both metabolism and meat fatty acids may affect the meat quality [24]. Thus, the present study investigated the interactive effects of inulin and Chinese gallotannin in the diets of Hu sheep, focusing specifically on the fatty acid composition of the meat and urine metabolites.

## 2. Materials and Methods

The animal experiments were conducted at Baiyin Kang Rui Sheep Breeding Co., Ltd., Gansu, China. All samples were gathered in strict accordance with the code of ethics approved by the Animal Welfare Committee of the College of Animal Science and Technology of Gansu Agricultural University (approval number: GSAU-2-th-AST-2021-001).

### 2.1. Experimental Animals and Sample Collection

Twenty-four healthy weaned and uncastrated Hu male lambs (approximately 4.5 months old) with similar body weights (25.80 ± 3.85 kg) were randomly selected from the lamb flock and divided into four groups using randomized block design: control group (basal diet, *n* = 6), treatment group I (basal diet + 0.1% inulin, *n* = 6), treatment group II (basal diet + 0.1% inulin + 2% Chinese gallotannin, *n* = 6), and treatment group III (basal diet + 0.1% inulin + 2% Chinese gallotannin + 4% PEG, *n* = 6). The weaning age for these lambs is around 2 months old. Inulin (purity > 95%) was purchased from Baiyin Rui Sheng Bioengineering Co., Ltd., Baiyin, China; Chinese gallotannin (purity 50–52%) was purchased from Qingdao RBT Biotechnology Co., Ltd., Qingdao, China; PEG was purchased from Tianjin BAISHI Chemical Industry Co., Ltd., Tianjin, China. All sheep were kept in metabolic cages (1.2 m × 1.5 m) individually, were fed twice a day (08:00 am and 17:00 pm), and accessed water freely. 

The study took 75 days including a 15-day adaptation period and a 60-day trail period. Daily feed intake was determined based on the difference between offered and residual feed to calculate the dry matter intake (DMI). The body weights (BW) of the individual sheep were measured monthly, and the total weight gain was calculated as the difference between the initial and final BWs. The feed conversion ratio (FCR) and average daily gain (ADG) were determined cumulatively through the collected data. At the end of the measurement period, a urine sample was collected for lipid metabolism analysis: urine samples were collected on the morning of 60td day of the feeding experiment, and the collection device was made according to Kowalczyk et al.’s (1996) description [25]. We transferred the urine to a 50 mL sterile cryopreservation tube immediately after urination Then, urine samples were stored at −80 °C for later analysis. The live weight (LW) of these sheep was determined before slaughter. Sheep were slaughtered humanely following the Islamic practice. The longissimus dorsi muscles were collected after slaughter and used to determine the fatty acid contents.

### 2.2. Basal Diet

The dietary formula was designed in accordance with the Chinese mutton sheep feeding standard (NY/T 816-2004), and the nutritional levels of the four groups were kept the same (Table 1).

### 2.3. Measurement of Muscle Fatty Acids

Muscle samples were preprocessed as previously described [26]. The surface fat of the meat samples was removed, and the samples were ground in a mortar with liquid nitrogen. One gram of the ground material was then placed in a 15 mL centrifuge tube, and 0.7 mL of KOH solution at a concentration of 10 mol/L and 5.3 mL of anhydrous methanol (analytical methanol, for chromatography) were added, respectively. The samples were placed in a thermostat water bath at 55 °C for 1.5 h, with shaking for 5 s per 20 min, and were then removed and cooled to below room temperature under tap water after the water bath finished. After this step, 0.58 mL of 12 mol/L H_2_S0_4_ solution was added, and the samples were again incubated in the water bath at 55 °C for 1.5 h for methyl esterification of the free fatty acids, with shaking for 5 s per 20 min. The supernatants were removed from the water bath and cooled to below room temperature under tap water, after which 3 mL of n-hexane was added to a centrifuge tube, and the material was centrifuged at 3000 rpm for 5 min. The supernatants were removed and filtered into a sample bottle using an organic phase filter membrane, and 2 mL of the sample was concentrated to below 1.5 mL at 45 °C. The processed samples were placed at −20 °C and analyzed for fatty acids by a gas chromatograph. We used an Agilent 6890 N gas chromatograph with an SP-2560 capillary column (100 m × 0.25 mm × 0.2μm, Sigma-Aldrich), FID hydrogen flame ion detector, carrier gas (N2) flow rate of 1.5 mL/min, and an air (combustion gas) flow rate of 400 mL/min. Each sample was injected in a split stream, with a split ratio of 1:100; the inlet temperature and the FID detector temperature were both 260 °C.

### 2.4. Extraction and LC-MS Analysis of Urine Metabolites

A pipette was used to draw 200 μL of defrosted urine into a 1.5 mL centrifuge tube, and 800 μL of precooled methanol/acetonitrile/water solution (4:4:2, *v*/*v*) was added. The mixture was swirled and then left to stand at −20 °C for 60 min; after centrifugation at 14,000 rpm 4 °C for 20 min, the supernatant was removed for vacuum drying. Mass spectrometric analysis was performed by adding 100 μL acetonitrile solution (acetonitrile: water = 1:1, *v*/*v*), redissolving, swirling, and centrifuging at 14,000 rpm for 15 min at 4 °C, after which the supernatant was fractionated by liquid chromatograph–mass spectrometer (LC-MS) analysis. A more detailed approach to the LC-MS procedure has been described in previous reports [27]. 

### 2.5. Statistical Analysis

SPSS 26 was used to analyze the meat fatty acids and production performance by using one-way ANOVA, and Duncan method was used for multiple comparisons. The data were presented as mean ± SD, and *p* < 0.05 was considered as statistically significant. GraphPad Prism 8.0.1 was used for graphics.

The raw data collected using MassLynx V4.2 were subjected to data processing operations, such as peak extraction and peak alignment by Progenesis QI software. The identification of the raw data was based on the online METLIN database of Progenesis QI software, with theoretical fragment identification and mass deviation within 100 ppm. After the original peak area information was normalized to the total peak area, a follow-up analysis was carried out. Principal component analysis and Spearman correlation analysis were used to judge the reproducibility of within-group samples and quality control samples. KEGG, HMDB, and lipidmaps databases were searched for the classification and pathway information on the identified compounds [28,29]. According to the grouping information, the difference multiples were calculated and compared, and the difference significance *p* value of each compound was calculated by a t-test. The R language package ropls was used to model OPLS-DA, and 200 permutation tests were conducted to verify the reliability of the model [30]. The differential multiples, *p* values, and VIP values of OPLS-DA model were used to screen the differential metabolites. The screening criteria were FC > 1, *p* < 0.05, and VIP > 1. Spearman correlations were used to analyze the relationships between muscle fatty acids and differential metabolites.

## 3. Results

### 3.1. Effect of Inulin and Chinese Gallotannin on Lamb Performance

There was no significant difference in ADG, DMI, FCR, and LW among groups (*p* > 0.05; Figure 1).

### 3.2. Effect of Inulin and Chinese Gallotannin on Meat Saturated Fatty Acids

Palmitic acid (C16:0) was the most abundant SFA and was found to be significantly lower in the treatment II group than in the other three groups (*p* < 0.05; Table 2; Figure 2 A). Myristic acid (C14:0) was also notably lower in treatment group II compared to the control group (*p* < 0.05; Table 2; Figure 2A). Stearic acid (C18:0) was also present at high levels, but these showed no obvious differences between the different groups (*p* > 0.05; Table 2; Figure 2A).

### 3.3. Effect of Inulin and Chinese Gallotannin on Meat Unsaturated Fatty Acid

The content of palmitoleic acid (C16:1) was greater in the treatment II group compared with the control group (*p* < 0.05; Table 3; Figure 2B), while elaidic acid (C18:1n9t) was more abundant in the treatment II group than in the other three groups (*p* < 0.05; Table 3; Figure 2B). Meanwhile, linoleic acid (C18:2n6c) was the highest among the 10 polyunsaturated fatty acids (PUFAs) in this study, with higher levels in the treatment group II and control groups than in the treatment I and treatment III groups (*p* < 0.05; Table 3; Figure 2C).

### 3.4. Effect of Inulin and Chinese Gallotannin on Meat Total Fatty Acid

The total SFA content in the treatment III group was higher than that in the other groups (*p* < 0.05; Table 4; Figure 2D). No difference was found among the groups for UFA, MUFA, and PUFA.

### 3.5. Screening of Differential Metabolites

The results of the OPLS-DA analysis showed that there were differences in the metabolites of Hu sheep between the control group and the three treatment groups (Figure 3A–C). Overall, we identified 146 differentially regulated metabolites (DRMs) between the control and treatment I groups, 467 DRMs between the control and treatment II groups, and 1399 DRMs between the control and treatment III groups (Figure 3D). In addition, we determined the number of common and unique DRMs in the four groups (Figure 3E). The results of the DRM clustering heatmap also confirmed considerable variations in the levels of urinary metabolites between the control group and the individual treatment groups (Figure 3F–H).

### 3.6. Functional Enrichment Analysis of Differential Metabolites

KEGG enrichment analysis of the DRMs among different groups presented the top 20 most significantly enriched metabolic pathways. The results confirmed that compared with the control group, the treatment I group exhibited enrichment in nicotinate and nicotinamide metabolism (ko00760); ether lipid metabolism (ko00565); vitamin B6 metabolism (ko00750); valine, leucine, and isoleucine biosynthesis (ko00290); and metabolic pathways (ko01100) (Figure 4A). The fatty acid biosynthesis pathway (ko00061) and ABC transporters pathway (ko02010) were more significantly enriched in the treatment II group (Figure 4B), while the glycerophospholipid metabolism (ko00564) and choline metabolism in cancer (ko05231) pathway were greater in the urine of the treatment III group (Figure 4C). At the same time, we selected the above metabolic pathways and DRMS, showing significant enrichment in the pathways for Sankey plot (Figure 4D). It can be clearly seen that the metabolic pathways involved by each DRMs, and some DRMs were also enriched in more than one metabolic pathway, such as l-threonine, 4-Pyridoxic acid, and PE (14:0/22:1(13Z)). Furthermore, Figure 4E–G also conforms to the observed upregulation and downregulation of DRMS in the control group and treatment groups I, II, and III.

### 3.7. Correlation Analysis

A correlation heatmap was created using Spearman correlations to identify possible relationships between the meat fatty acids and urine metabolites (Figure 5). The consequences confirmed that most of the muscle fatty acids were associated with urinary metabolites. Creatinine was found to be positively correlated with monounsaturated fatty acid (MUFA), and C18:1n9t and was negatively correlated with SFA and C17:0. Nicotinuric acid was positively correlated with C18:2n6c and negatively correlated with various fatty acids, such as C16:0, while 4-Pyridoxic acid was positively correlated with MUFA and C18:1n9t and negatively correlated with other fatty acids. In addition, muscle fatty acids that were positively correlated with LysoPC (16:0), including C14:0, C16:0, C18:1n9t, C18:2n6c, and SFA, had negative correlations with MUFA and C17:0. PE (14:0/24:0) was negatively associated with C18:2n6c and C17:0, but was positively associated with all other muscle fatty acids.

## 4. Discussion

### 4.1. Analysis of Meat Fatty Acid

Meat fatty acids are related to both meat tenderness and the formation of flavor compounds [31]. Of these, n-3 PUFA and n-6 PUFA are known as functional fatty acids for their anticancer, lipid-lowering, and cardiovascular disease-preventing properties [32,33]. In the current study, compared with the control group, the treatment II group showed lower SFA and increased UFA contents. Previous research has found that while the presence of excessive SFAs in foods does not solely lead to elevated total cholesterol (TCH) in the serum and low-density lipoprotein (LDL) levels, but it may also lead to cardiovascular disease [34]. Moreover, there is a positive correlation between C14:0 and C16:0 in SFA and serum TCH levels. Increases in these levels may lead to an elevation of LDL levels in the blood, which is a risk factor for cardiovascular disease [35]. It is known that UFA (consisting of PUFA and MUFA) is essential in the prevention of cardiovascular disorders [36,37]. As a UFA, C18:2n6c does not only improve meat quality, but can also lower blood triglyceride and cholesterol concentrations, with anticancer, and antidiabetic properties together with lowering the risk of cardiovascular disease [38]. The reduced contents of C14:0 and C16:0 and the increased content of C18:2n6c in the treatment II group indicate that the combination of inulin and Chinese gallotannin may be beneficial for the improvement of mutton quality.

### 4.2. Effect of Inulin Addition to Diet on Urine Metabolism in Hu Sheep

As an important prebiotic, inulin has beneficial impacts in both humans and animals [39,40,41]. In the present study, the addition of 0.1% inulin to the diet significantly changed the contents of nicotinuric acid, 4-pyridoxic acid, N1-Methyl-2-pyridone-5-carboxamide, and l-threonine in the urine of Hu sheep. It was reported that nicotinic acid could be derived from tryptophan and mainly involved in redox reactions as a core enzyme component as well as providing energy [42]. Furthermore, nicotinic acid is also an acyl glycine, acting as a coenzyme in several biological processes in the body, including glycolysis, gluconeogenesis, the citric acid cycle, and oxidation, as well as the formation of long-chain fatty acids [43]. Besides, the presence of acyl glycine in the urine could reflect the accumulation of acyl-CoA ester in the mitochondria [44]. Meanwhile, long-chain acyl-CoA ester, a major indicator of muscle lipid metabolism, may be negatively correlated with the effects of insulin [45,46]. Thus, changes in lipid metabolism may also affect blood glucose content. Moreover, l-threonine is mainly involved in the biosynthesis of amino acids and the metabolism of glycine, serine, and threonine in animals and plays a role in the regulation of energy homeostasis, nutrient metabolism, intestinal health, immunity, and ailment [47]. Thus, the addition of inulin may adjust the sugar metabolism and lipid metabolism of sheep by interfering with the metabolism of nicotinic acid and l-threonine, which are metabolites concerned with the metabolism of nicotinate and amino acids.

### 4.3. Effect of Inulin and Chinese Gallotannin Addition to Diet on Urine Metabolism in Hu Sheep

To clarify the potential mechanism responsible for the effects of the combination of inulin and Chinese gallotannin, the fatty acid biosynthesis pathway and ABC transporters pathway were investigated. We found that after the combined addition of inulin and Chinese gallotannin to the diet, the proportions of creatinine and 4-pyridoxic acid in the urine of Hu sheep increased, while the levels of palmitic acid, oleic acid, and other metabolites were downregulated. Research has found that creatinine is present in constant levels in the body and mostly filtered by way of the glomerulus without being absorbed by the renal tubules, so that changes in the creatinine levels primarily mirror the glomerular filtration function (renal excretory capacity) [48,49]. Thus, the increase in the urinary creatinine content in treatment group II suggests that the addition of inulin and Chinese gallotannin may have beneficial effects on the excretory capacity of the kidneys. Furthermore, 4-pyridoxic acid is a significant metabolite of vitamin B6 and can also be used as an indicator of amino acid metabolism in animals when amino acids are excreted via the kidneys and urine [50,51]. It has been reported that elevated levels of palmitic and oleic acid in animals can lead to pancreatic β-cell dysfunction and cardiomyocyte apoptosis, resulting in abnormal lipid metabolism [52,53]. Furthermore, the palmitic and oleic acid levels are related to changes in body weight [54]. However, in the current study, the contents of both palmitic and oleic acid were substantially decreased in the treatment II group compared with the control group, suggesting that the combination of inulin and Chinese gallotannin may regulate dysfunctional lipid metabolism in Hu sheep through modulating the metabolic and synthetic pathways of fatty acids, such as palmitic and oleic acids, ultimately enhancing the mutton quality.

### 4.4. Effect of Inulin, Chinese Gallotannin, and PEG Addition to Diet on Urine Metabolism in Hu Sheep

These three substances (inulin, Chinese gallotannin, and PEG) may be not suitable for simultaneous use as feed additives. We found that Lysophosphatidylcholine (LysoPC(20:5(5Z,8Z,11Z,14Z,17Z)), LysoPC(14:1(9Z)), LysoPC (16:0), and other LPCs in the urine of Hu sheep and phosphatidylethanolamine (PE (14:0/22:1(13Z)), PE (14:0/24:0)) were significantly raised in treatment III group (Figure 4G). These DRMs are significantly associated with glycerophospholipid and choline metabolism in the cancer associated with the sphingolipid signaling pathway (Figure 4D). Previous studies have shown that PE is one of the components of biological membranes, where it functions as a molecular chaperone, assisting in the correct folding of several membrane-associated proteins [55]. Moreover, LPC is an essential component of oxidized low-density lipoprotein, which is known to increase the production of inflammatory cytokines and has a variety of biological functions in cardiovascular disease, as well as being used as a biomarker for several diseases [56,57]. LPC has been shown to promote the formation of atherosclerotic plaques, exacerbate inflammation, enhance anti-infective responses, modify blood glucose, and also influence tumor invasion and metastasis [58,59]. It is thus concluded that the combination of inulin, Chinese gallotannin, and PEG may stimulate inflammatory reactions and lipid metabolism disorders in Hu sheep, which may be not conducive to the health of the sheep.

### 4.5. Correlation Analysis

The correlation analysis between meat fatty acids and urine metabolism indicated that these were clearly related and may influence one another, which give us a way to better understand the mechanism of the effect of additives on animal production performance. For example, 4-pyridoxic acid correlated with muscle fatty acids, which is a metabolite of vitamin B6. Vitamin B6 contributes to fatty acid biosynthesis [60]. This suggests that part of the action mechanism of inulin and Chinese gallotannin is to affect the synthesis of vitamin B6, and then affect the synthesis of fatty acids in animals. This is consistent with the fact that one of the nutritional effects of inulin is to benefit the synthesis of B vitamins by stimulating bifidobacteria growth in the intestine [61]. However, its specific mechanism needs to be further studied.

### 4.6. The Clinical Significance of the Study

Maintaining the balanced lipid metabolism is essential for both humans and animals. Lipids, as structural elements of cell membranes, could influence metabolic homeostasis through a variety of mechanisms and are necessary for the maintenance of metabolic homeostasis. As a necessary category of lipids, sphingolipids play vital roles in organisms. Complex sphingolipids can be converted into biologically energetic metabolites, such as sphingosine, sphingomyelin, and ceramide, that play essential roles in promoting cell recognition; inhibiting proliferation; and stimulating apoptosis, signal transduction, and cytokinesis [62,63]. Besides, the disordered lipid metabolism is also one of the essential elements influencing metabolic diseases [64,65]. In this study, the sheep in treatment group III showed some inflammatory reactions resulting from dysregulated lipid metabolism, which could be harmful to the health of the animal. Thus, the prevention of lipid metabolism disorders is necessary for both humans and animals.

Fatty acids play central roles in growth and development through their roles in membrane lipids, as ligands for receptors and transcription factors that regulate gene expression, as precursors for eicosanoids, in cellular communication, and through direct interactions with proteins [66]. The addition of inulin and Chinese gallotannin in diet would change the content of fatty acids in sheep. A lower the content of SFA and a higher the content of UFA is good for sheep health, which is in line with the demand of producers and consumers.

## 5. Conclusions

The combination of inulin and Chinese gallotannin (basal diet + 0.1% inulin + 2% Chinese gallotannin) as diet additives was beneficial in reducing the content of SFA and increasing the content of UFA in mutton, while enhancing both energy and lipid metabolism in Hu sheep. However, the combined addition of inulin, Chinese gallotannin, and PEG (basal diet + 0.1% inulin + 2% Chinese gallotannin + 4% PEG) under the current experimental situation did not produce a beneficial effect on the reduction of SFA content in mutton and is likely to cause inflammatory reactions and lipid metabolism disorders in Hu sheep.

## Figures and Tables

**Figure 1 animals-13-00160-f001:**
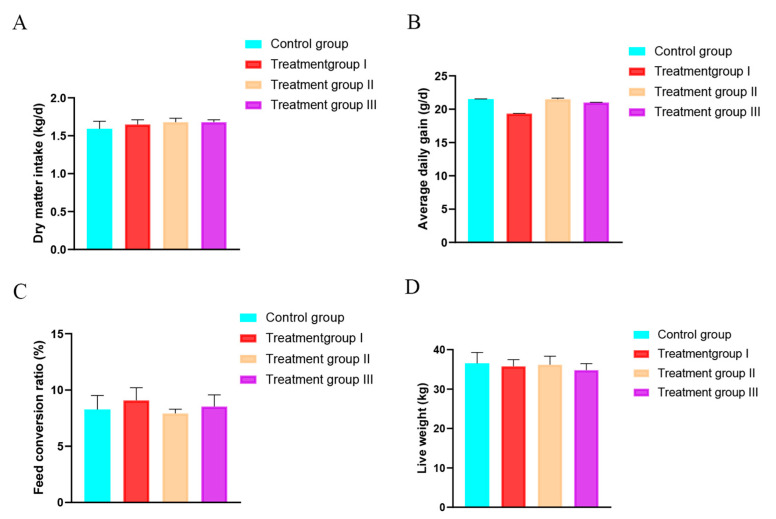
Effect of inulin and Chinese gallotannin on growth performance in sheep. (**A**) Dry matter intake; (**B**) average daily gain; (**C**) feed conversion ratio; (**D**) live weight.

**Figure 2 animals-13-00160-f002:**
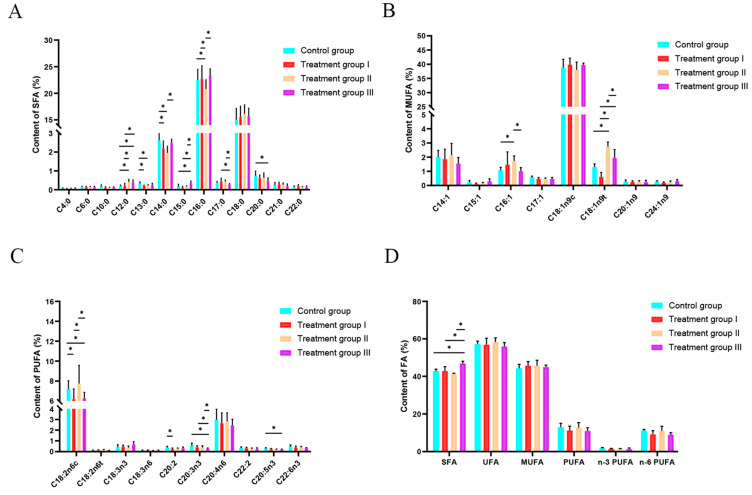
Analysis of the composition and contents of various fatty acids in the muscle tissues of lambs from the control and treatment groups I, II and III. (**A**) Composition and contents of saturated fatty acids; (**B**) composition and contents of monounsaturated fatty acids; (**C**) composition and contents of polyunsaturated fatty acids; (**D**) composition and contents of total fatty acids. * Indicates a significant difference between different trial groups (*p* < 0.05).

**Figure 3 animals-13-00160-f003:**
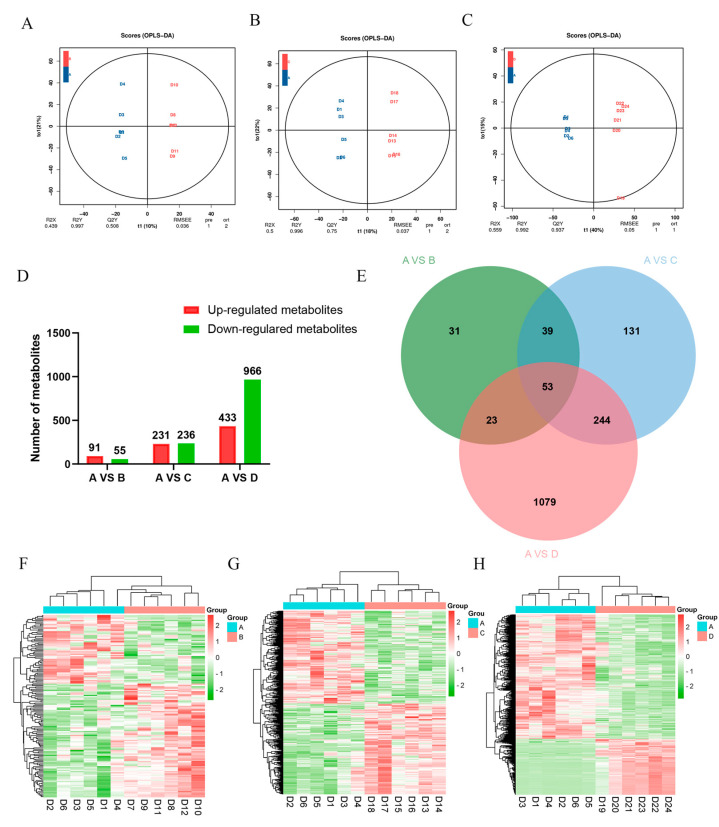
Analysis of differential metabolites between the control group and treatment groups I, II, and III of Hu sheep urine. (**A**–**C**) Clustering plots of partial least squares discriminant analysis (PLS-DA); (**D**) differentially regulated metabolites between the control group and treatment groups I, II, and III; (**E**) Venn diagram of differential metabolites between the different groups; (**F**–**H**) cluster heatmap of the differential metabolites between the control group and the treatment groups I, II, and III. **A**, control group (basal diet); **B**, treatment group I (basal diet + 0.1% inulin); **C**, treatment group II (basal diet + 0.1% inulin + 2% Chinese gallotannin); **D**, treatment group III (basal diet + 0.1% inulin + 2% Chinese gallotannin + 4% PEG).

**Figure 4 animals-13-00160-f004:**
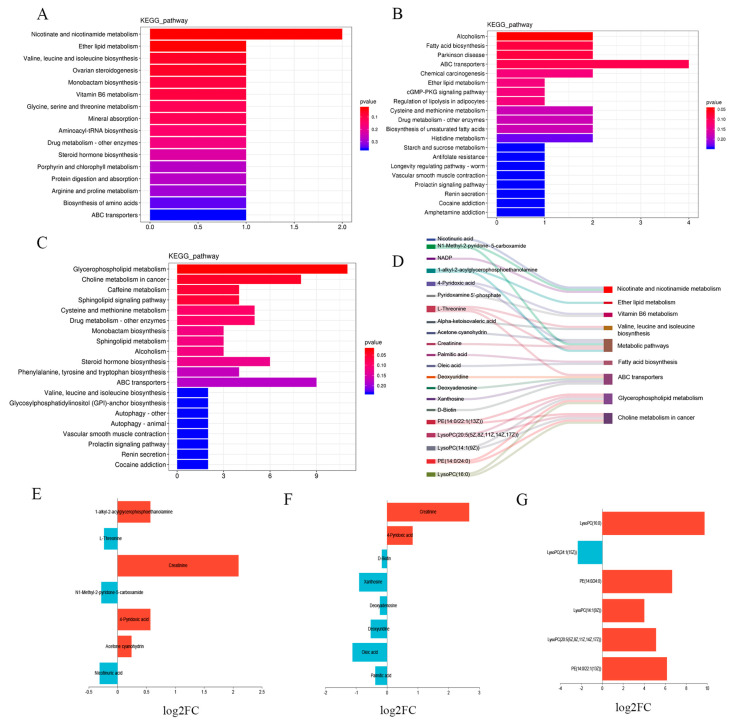
Functional enrichment analysis of differential metabolites in the urine of the control and treatment groups I, II, and III in Hu sheep. (**A**) Bar graphs of KEGG functional enrichment in control and treatment I groups; (**B**) bar graphs of KEGG functional enrichment in the control and treatment II groups; (**C**) bar graphs of KEGG functional enrichment in the control and treatment III groups; (**D**) graphs of differential metabolites in relation to their enriched metabolic pathways; (**E**) urinary differential metabolites between the control and treatment I groups; (**F**) urinary differential metabolites between the control and treatment II groups; (**G**) urinary differential metabolites in the control group versus the treatment III group; control group, basal diet; treatment group I = basal diet + 0.1% inulin; treatment group II, basal diet + 0.1% inulin + 2% Chinese gallotannin; treatment group III, basal diet + 0.1% inulin + 2% Chinese gallotannin + 4% PEG.

**Figure 5 animals-13-00160-f005:**
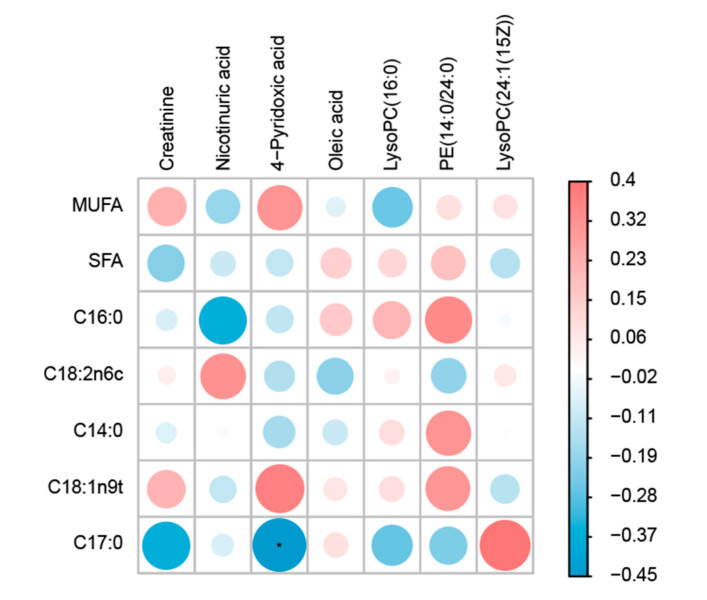
Spearman correlations between fatty acids and urine differential metabolites of sheep.

**Table 1 animals-13-00160-t001:** Basic diet composition and nutrient levels (dry matter basis).

Raw Material	Proportion	Nutrient Levels	Content
Ensilage corn stalks (%)	8	DM (kg/d)	1
Alfalfa hay (%)	22	ME (MJ/kg)	12.01
Corn (%)	34	CP (%)	18.17
Soybean meal (%)	14	Ca (%)	0.31
Rice bran (%)	3	P (%)	0.30
Flax cake (%)	3	NDF (%)	23.22
Corn DDGS (%)	12		
NaCl (%)	0.7		
Premix ^1^ (%)	0.5		
Puffing urea (%)	1.3		
NaHCO_3_ (%)	1.5		
Total (%)	100		

^1^ Premix is supplied per kilogram of feed: Vitamin A, 400–1000 IU; Vitamin D3, 75–400 IU; Vitamin E, 0.6–10 mg; Iodine, 0.075–0.425 mg; Selenium, 0.015–0.05 mg; Cobalt, 0.05–0.20 mg; Manganese, 4.1–15 mg; Fe, 4.5–18.08 mg; Zinc, 7.5–12 mg; Copper, 0.35–1.4 mg; Ca, 0.3 -1.2 mg; P, 0.075–2.5 mg.

**Table 2 animals-13-00160-t002:** Effect of inulin and Chinese gallotannin on meat saturated fatty acids in sheep.

Item	Treatment				SEM	*p*-Value
	A ^1^	B ^2^	C ^3^	D ^4^		
C4:0 (%)	0.08 ± 0.02	0.05 ± 0.02	0.06 ± 0.03	0.06 ± 0.04	0.01	0.271
C6:0 (%)	0.15 ± 0.03	0.15 ± 0.03	0.13 ± 0.04	0.15 ± 0.03	0.01	0.630
C10:0 (%)	0.17 ± 0.09	0.12 ± 0.04	0.11 ± 0.02	0.14 ± 0.02	0.01	0.277
C12:0 (%)	0.20 ^b^ ± 0.05	0.22 ^b^ ± 0.12	0.41 ^a^ ± 0.14	0.39 ^a^ ± 0.13	0.03	0.006
C13:0 (%)	0.33 ^a^ ± 0.09	0.21 ^b^ ± 0.03	0.22 ^b^ ± 0.05	0.25 ^ab^ ± 0.09	0.02	0.036
C14:0 (%)	2.62 ^a^ ± 0.88	2.19 ^bc^ ± 0.39	1.95 ^c^ ± 0.35	2.49 ^ab^ ± 0.19	0.11	0.001
C15:0 (%)	0.17 ^b^ ± 0.12	0.11 ^b^ ± 0.07	0.16 ^b^ ± 0.05	0.35 ^a^ ± 0.11	0.03	0.001
C16:0 (%)	22.53 ^a^ ± 1.98	22.77 ^a^ ± 2.40	20.79 ^b^ ±1.73	23.41 ^a^ ± 1.20	0.41	0.002
C17:0 (%)	0.37 ^ab^ ± 0.07	0.44 ^a^ ± 0.18	0.43 ^a^ ± 0.09	0.28 ^b^ ± 0.05	0.03	0.042
C18:0 (%)	15.03 ± 2.13	15.52 ± 2.02	16.13 ± 1.70	15.65 ± 1.55	0.36	0.788
C20:0 (%)	0.74 ^a^ ± 0.25	0.62 ^ab^ ± 0.15	0.63 ^ab^ ± 0.26	0.46 ^b^ ± 0.17	0.05	0.019
C21:0 (%)	0.25 ± 0.14	0.31 ± 0.09	0.26 ± 0.10	0.19 ± 0.12	0.03	0.418
C22:0 (%)	0.16 ± 0.05	0.20 ± 0.07	0.13 ± 0.04	0.16 ± 0.05	0.01	0.272

C4:0, butyric acid; C6:0, hexanoic acid; C10:0, decanoic acid; C12:0, lauric acid; C13:0, tridecanoic acid; C14:0, myristic acid; C15:0, pentadecanoic acid; C16:0, palmitic acid; C17:0, heptadecanoic acid; C18:0, stearic acid; C20:0, arachidic acid; C21:0, heneicosanoic acid; C22:0, behenic acid; PEG, polyethylene glycol; ^1^ control group (basal diet); ^2^ treatment group I (basal diet + 0.1% inulin); ^3^ treatment group II (basal diet + 0.1% inulin + 2% Chinese gallotannin); ^4^ treatment group III (basal diet + 0.1% inulin + 2% Chinese gallotannin +4% PEG); a–c Values within a row with different superscripts differ significantly at *p* < 0.05.

**Table 3 animals-13-00160-t003:** Effect of inulin and Chinese gallotannin on meat unsaturated fatty acids in sheep.

Item	Treatment				SEM	*p*-Value
	A ^1^	B ^2^	C ^3^	D ^4^		
MUFA						
C14:1 (%)	2.01 ± 0.46	1.87 ± 0.69	2.14 ± 0.83	1.55 ± 0.42	0.12	0.409
C15:1 (%)	0.23 ± 0.12	0.11 ± 0.06	0.15 ± 0.06	0.29 ± 0.18	0.03	0.073
C16:1 (%)	1.08 ^b^ ± 0.20	1.47 ^ab^ ± 0.90	1.72 ^a^ ± 0.37	1.01 ^b^ ± 0.25	0.12	0.045
C17:1 (%)	0.57 ± 0.09	0.47 ± 0.09	0.47 ± 0.06	0.47 ± 0.10	0.02	0.151
C18:1n9c (%)	38.83 ± 2.90	39.86 ± 2.26	37.96 ± 2.76	39.75 ± 0.64	0.47	0.465
C18:1n9t (%)	1.31 ^bc^ ± 0.21	0.61 ^c^ ± 0.31	2.73 ^a^ ± 0.33	1.94 ^b^ ± 0.59	0.22	0.001
C20:1n9 (%)	0.26 ± 0.13	0.25 ± 0.08	0.26 ± 0.08	0.24 ± 0.13	0.02	0.987
C24:1n9 (%)	0.26 ± 0.07	0.19 ± 0.06	0.23 ± 0.09	0.29 ± 0.12	0.02	0.341
PUFA						
C18:2n6c (%)	7.21 ^a^ ± 0.82	6.20 ^b^ ± 1.00	7.75 ^a^ ± 1.82	6.27 ^b^ ± 0.60	0.26	0.034
C18:2n6t (%)	0.11 ± 0.04	0.12 ± 0.05	0.14 ± 0.07	0.09 ± 0.03	0.01	0.431
C18:3n3 (%)	0.43 ± 0.20	0.45 ± 0.14	0.38 ± 0.15	0.65 ± 0.27	0.05	0.258
C18:3n6 (%)	0.12 ± 0.03	0.13 ± 0.02	0.09 ± 0.02	0.10 ± 0.05	0.01	0.401
C20:2 (%)	0.39 ^a^ ± 0.11	0.23 ^b^ ± 0.11	0.27 ^ab^ ± 0.07	0.30 ^ab^ ± 0.14	0.02	0.028
C20:3n3 (%)	0.55 ^a^ ± 0.24	0.40 ^a^ ± 0.14	0.39 ^a^ ± 0.15	0.28 ^b^ ± 0.07	0.04	0.026
C20:4n6 (%)	2.97 ± 1.00	2.65 ± 0.97	2.81 ± 0.80	2.47 ± 0.55	0.17	0.771
C22:2 (%)	0.34 ± 0.11	0.33 ± 0.08	0.26 ± 0.09	0.24 ± 0.12	0.02	0.279
C20:5n3 (%)	0.29 ^a^ ± 0.08	0.24 ^ab^ ± 0.06	0.21 ^ab^ ± 0.03	0.19 ^b^ ± 0.05	0.02	0.038
C22:6n3 (%)	0.48 ± 0.13	0.39 ± 0.13	0.40 ± 0.09	0.34 ± 0.04	0.02	0.280

MUFA, monounsaturated fatty acid; C14:1, myristoleic acid; C15:1, cis-10-Pentadecenoic acid; C16:1, palmitoleic acid; C17:1, cis-10-Heptadecenoicacid; C18:1n9c, oleic acid; C18:1n9t, elaidic acid; C20:1n9, cis-11-Eicosenoic acid; C24:1n9, nervonic acid; PUFA, polyunsaturated fatty acid; C18:2n6c, linoleic acid; C18:2n6t, linolelaidic acid; C18:3n3, α-linolenic acid; C18:3n6, γ-linolenic acid; C20:2, cis-11,14-Eicosatrienoic acid; C20:3n3, cis-11,14,17-Eicosatrienoic acid; C20:4n6, arachidonic acid; C22:2, cis-13,16-Docosadienoic acid; C20:5n3, cis-5,8,11,14,17-Eicosapentaenoic acid; C22:6n3, cis-4,7,10,13,16,19-Docosahexaenoic acid; PEG, polyethylene glycol; ^1^ control group (basal diet); ^2^ treatment group I (basal diet + 0.1% inulin); ^3^ treatment group II (basal diet + 0.1% inulin + 2% Chinese gallotannin); ^4^ treatment group III (basal diet +0.1% inulin + 2% Chinese gallotannin + 4% PEG); a–c Values within a row with different superscripts differ significantly at *p* < 0.05.

**Table 4 animals-13-00160-t004:** Effect of inulin and Chinese gallotannin on total fatty acids in sheep meat.

Item	Treatment				SEM	*p*-Value
	A ^1^	B ^2^	C ^3^	D ^4^		
SFA (%)	42.92 ^b^ ± 0.90	42.91 ^b^ ± 2.27	41.27 ^b^ ± 0.44	46.91 ^a^ ± 1.15	0.62	0.001
UFA (%)	57.16 ± 1.62	56.88 ± 3.45	58.29 ± 2.20	55.93 ± 2.07	0.49	0.431
MUFA (%)	44.27 ± 2.14	45.74 ± 2.10	45.64 ± 2.98	44.98 ± 1.04	0.43	0.268
PUFA (%)	12.89 ± 2.11	11.14 ± 2.28	12.66 ± 2.71	10.96 ± 1.67	0.46	0.327
n-3PUFA (%)	1.76 ± 0.38	1.48 ± 0.26	1.32 ± 0.36	1.36 ± 0.59	0.09	0.239
n-6PUFA (%)	11.10 ± 0.67	9.10 ± 1.95	10.80 ± 2.55	8.92 ± 1.08	0.40	0.104

SFA, saturated fatty acids; UFA, unsaturated fatty acid; MUFA, monounsaturated fatty acid; PUFA, polyunsaturated fatty acid, PEG, polyethylene glycol; n-3PUFA, n-3polyunsaturated fatty acid (C18:3n3, C20:3n3, C20:5n3, C22:6n3); n-6PUFA, n-6polyunsaturated fatty acid (C18:2n6c, C18:2n6t, C18:3n6, C20:4n6); ^1^ control group (basal diet); ^2^ treatment group I (basal diet + 0.1% inulin); ^3^ treatment group II (basal diet + 0.1% inulin + 2% Chinese gallotannin); ^4^ treatment group III (basal diet + 0.1% inulin + 2% Chinese gallotannin + 4% PEG); a–b Values within a row with different superscripts differ significantly at *p* < 0.05.

## Data Availability

None of the data were deposited in an official repository.

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
