# Peer review of "Inulin and Chinese Gallotannin Affect Meat Quality and Lipid Metabolism on Hu Sheep"

_animals, 2022, doi:10.3390/ani13010160_

Round 1
Reviewer 1 Report
Introduction
-This section of the manuscript needs to be extended.
-Please extend the text about inulin and gallotanin to allow readers of the paper a better understanding.
Materials and methods
-Please provide details of selection of animals and the method for allocation into groups.
-How were the animals euthanised?
-How was urine collected?
Results
-Tables 3, 4, 5 are too detailed and difficult to read. Please transfer to supplementary material and provide only a brief summary in text.
Discussion
-A lot of the text in Discussion is really results and must be transferred to the correct section.
Author Response
Response to Reviewer 1:
Dear Reviewer:
Thank you for your review and suggestions on our manuscript. Those comments are all valuable and very helpful for revising and improving our paper. We have studied comments carefully and have made correction which we hope meet with approval.
The main corrections in the paper and the responds to your comments are as flowing:
Points1: Manuscript should undergo extensive English revisions.
Response: We have carefully modified the manuscript, and then it has been polished by MJ language editing services, Shenzhen, China.
Point 2: Please extend the text about inulin and gallotanin to allow readers of the paper a better understanding
Response: Thank you for carefully reading our manuscript and giving us valuable suggestion. We have added the content about inulin (line: 46-49) and gallotanin (line: 60-64) in the introduction section.
Point 3: Please provide details of selection of animals and the method for allocation into groups.
Response: Thank you for your review and suggestions. We have added the details of selection of animals and the method for allocation into groups in manuscript (line: 92-104). The details are as follows:
Twenty-four healthy weaned Hu sheep male lambs (approximately 4.5 months old) with similar body weights (25.80 ± 3.85 kg) were randomly selected from lambs flock and divided into 4 four groups using randomized block design: control group (basal diet, n = 6), treatment group I (basal diet + 0.1% inulin, n = 6), treatment group II (basal diet + 0.1% inulin + 2% Chinese gallotannin, n = 6), treatment group III (basal diet + 0.1% inu-lin + 2% Chinese gallotannin + 4% PEG, n = 6).
Point 4: How were the animals euthanised?
Response:
The detail was added in “2.1. Experimental animals and sample collection”. All lambs were slaughtered humanely following the Islamic practice.
Point 5: How was urine collected?
Response: As sheep were kept in metabolic cages which allowed us to collect urine conveniently. In addition, feces collection bags were attached to the sheep buttocks which prevent feces from polluting urine.
Point 6: Tables 3, 4, 5 are too detailed and difficult to read. Please transfer to supplementary material and provide only a brief summary in text.
Response: In order to facilitate readers to better read the relevant content, we made a histogram (Figure 2) which could briefly show the changes of meat fatty acids among groups. But we still choose to leave Tables 3, 4, 5 in manuscript as for the muscle fatty acids analysis is a main part of this study.
Point 7: Discussion -A lot of the text in Discussion is really results and must be transferred to the correct section.
Response: Thank you for your review and suggestions, we have deleted relevant contents in the discussion accordingly and made further improvements.

Reviewer 2 Report
Introduction
The hypothesis of the authors is missing. Also, the objectives of the study must be presented clearly.
M&M
Please provide age of animals. Also, were lambs castrated or entire?
The 15 day pre-trial adaptation is NOT in the trial, so the trial was only 60-day long, please correct.
How did you confirm that results had a normal distribution?
Results
Additionally to table 2, please provide a graph with the same results in visual presentation.
Discussion
Some recent (2022) relevant references with similar work are missing and must be cited and discussed.
Author Response
Response to Reviewer 2
Dear Reviewer:
Thank you for your review and suggestions on our manuscript. Those comments are all valuable and very helpful for revising and improving our paper. We have studied comments carefully and have made correction which we hope meet with approval.
The main corrections in the paper and the responds to your comments are as flowing:
Point 1: Introduction- The hypothesis of the authors is missing. Also, the objectives of the study must be presented clearly.
Response: Thank you for carefully reading our manuscript and giving us valuable suggestion. We have perfected the objectives and hypothesis (line:78-85) of the study. The details are as follows:
Line 78-85: Since inulin is known to interact beneficially with other feed additives for livestock, we speculated that there may be an interactive and beneficial effect between inulin and Chinese gallotannin on animal health and production as both metabolism and meat fatty acids may affect the meat quality. Thus, the present study investigated the interactive effects of inulin and Chinese gallotannin in the diets of Hu sheep, focusing specifically on the fatty acid composition of the meat and urine metabolites.
Point 2: Please provide age of animals. Also, were lambs castrated or entire?
Response: These details added in the manuscript. “Twenty-four healthy weaned and uncastrated Hu male lambs (approximately 4.5 months old)…¼”
Point 3: The 15 day pre-trial adaptation is NOT in the trial, so the trial was only 60-day long, please correct.
Response: Done.
Point 4: How did you confirm that results had a normal distribution?
Response: We use the skewness and kurtosis of the data for normality test. In this study, the skewness and kurtosis of the data are within a reasonable range (-1.96 – 1.96).
,
Point 5: Results- Additionally to table 2, please provide a graph with the same results in visual presentation.
Response: Thank you for carefully reading our manuscript and giving us valuable suggestion. We made a histogram (line 192, Figure 1) to show the data, which could clearly show the change among groups. The details are as follows:
Point 6: Some recent (2022) relevant references with similar work are missing and must be cited and discussed.
Response: We have added some latest relevant literatures in the discussion section. The details as follows:
Line 322-324: Muscle fatty acids are related to both meat tenderness and the formation of flavour compounds [2]. Of these n-3 PUFA and n-6 PUFA are known as functional fatty acids for their anti-cancer, lipid-lowering and cardiovascular-disease-preventing properties [3,4].
Line 344-345: As an important prebiotic, inulin has beneficial impacts in both humans and animals [5-7].
Line 383-386: It has been reported that elevated levels of palmitic and oleic acid in animals can lead to pancreatic β - cell dysfunction and cardiomyocyte apoptosis, resulting in abnormal lipid metabolism [8,9]. Furthermore, the palmitic and oleic acid levels are related to changes in body weight [10].
Line 422-425: Complex sphingolipids can be converted into biologically energetic metabolites such as sphingosine, sphingomyelin and ceramide that play essential roles in promoting cell recognition, inhibiting proliferation, and stimulating apoptosis, signal transduction and cytokinesis [11,12]. Besides, the disordered lipid metabolism is also one of the essential elements influencing metabolic diseases [13,14].

Reviewer 3 Report
Dear Editor and Authors,
I have reviewed the manuscript entitled "Inulin and Chinese gallotannin affect meat quality and lipid metabolism on Hu sheep" (animals-2096155). The subject of the study falls within the scope of the journal. An adequate amount of information has been provided in the introduction, and the researchers have clearly outlined their objectives. There is a clear description of the authors' materials and the methods they employed in their study. There is sufficient information presented in the results and discussion sections. There are, however, a few points that need to be improved.
Major comments:
There should be consistency between the naming of the trial groups in the abstract and material method section (control , group I , group II , group III )and in the result section (A, B, C, D).
L159, 167, 181, 200, 211, 231, 258: There should be appropriate subheadings in the result section. You may use "Effect of inulin and Chinese gallotannin on lamb performance" as an example.
Figure 3 provides a limited number of correlations between meat fatty acids and urine metabolites, despite the fact that more meat fatty acids and urine metabolites were detected in the study. The situation needs to be clarified.
It is unusual for the authors to provide a detailed description of the impact of each group. This is an excellent achievement for them, and I congratulate them.However, this is not apparent in the discussion of the correlation results.
Minor comments:
L13: Please use another expression for “ruminantia farming”.
L20-L79: It is necessary to specify the weaning age
L34: Please use “energy and lipid metabolism” instead of “energy metabolism and lipid metabolism”.
L49: Please italicize “Bacillus subtilis”.
L84: Please use “PEG” instead of “Polyethylene glycol”.
L85: Please use “induvial” instead of “single”.
L86: Please use “and accessed” instead of ”accessed”
L126: Please use “1:100” instead of ”100:1”
L132, 134: Please use “rpm” instead of ”r”
L134: Please use “v/v” instead of ”V/V”
e.g., L152, 156, 160, 189: It would be helpful if the P value were used consistently throughout the document
L259: Please use “Spearman” instead of ” spearman”.
Author Response
Response to Reviewer 3:
Thank you for your review and suggestions on our manuscript. We have studied comments carefully and have made correction which we hope meet with approval.
Point 1: There should be consistency between the naming of the trial groups in the abstract and material method section (control , group I , group II , group III )and in the result section (A, B, C, D).
Response: Thank you for carefully reading our manuscript and giving us valuable suggestion. This opinion is very important, and consistent description is really more convenient to understand the context. In the results section, we use A1, B2, C3 and D4 to represent the control group, treatment group I, treatment group II, and treatment group III respectively in Table 2, 3, 4 and 5, because these headings (control group, treatment group I, treatment group II, and treatment group III) are too long to be displayed in the same row if they are placed in the table. Therefore, we choose to use A1. B2, C3 and D4 respectively to replace. In addition, we have also made corresponding comments below each table such as:
1 control group (basal diet); 2 treatment group I (basal diet + 0.1% inulin); 3 treatment group II (basal diet + 0.1% inulin + 2% Chinese gallotannin); 4 treatment group III (basal diet + 0.1% inulin + 2% Chinese gallotannin + 4% PEG).
Point 2: L159, 167, 181, 200, 211, 231, 258: There should be appropriate subheadings in the result section. You may use "Effect of inulin and Chinese gallotannin on lamb performance" as an example.
Response: Thank you for carefully reading our manuscript and giving us valuable suggestion. We have revised the above title accordingly.
Point 3: Figure 3 provides a limited number of correlations between meat fatty acids and urine metabolites, despite the fact that more meat fatty acids and urine metabolites were detected in the study. The situation needs to be clarified.
Response: We selected some muscle fatty acids with high percentage content and important differential regulation metabolites screened by urine metabolomics to explore and analyze the correlation between them. Therefore, only a limited number of correlations are shown in Figure 3.
Point 4: It is unusual for the authors to provide a detailed description of the impact of each group. This is an excellent achievement for them, and I congratulate them. However, this is not apparent in the discussion of the correlation results.
Response: Thank you for carefully reading our manuscript and giving us valuable suggestion. We have added some discussions about it.
Point 5: L13: Please use another expression for “ruminantia farming”.
Response: Done
Point 6: L20-L79: It is necessary to specify the weaning age
Response: We have added it into the manuscript. “The weaning age for these lambs is around 2 months old.”
Point 7: L34: Please use “energy and lipid metabolism” instead of “energy metabolism and lipid metabolism”.
Response: Done
Point 8: L49: Please italicize “Bacillus subtilis”.
Response: Done
Point 9: L84: Please use “PEG” instead of “Polyethylene glycol”.
Response: Done
Point 10: L85: Please use “induvial” instead of “single”.
Response: Done
Point 11: L86: Please use “and accessed” instead of ”accessed”.
Response: Done
Point 12: L126: Please use “1:100” instead of ”100:1”.
Response: Done
Point 13: L132, 134: Please use “rpm” instead of ”r”.
Response: Done
Point 14: L134: Please use “v/v” instead of ”V/V”.
Response: Done
Point 15: e.g., L152, 156, 160, 189: It would be helpful if the P value were used consistently throughout the document.
Response: Done
Point 16: L259: Please use “Spearman” instead of “spearman”.
Response: Done

Round 2
Reviewer 1 Report
The authors have NOT clarified how urine was collected.
The answer is vague and does not explain the procedure followed. There are various methods for urine collection from sheep, so the method employed must be described clearly.
Author Response
Response to Reviewer 1:
Dear Reviewer:
Thank you for your review and suggestions on our manuscript. The main corrections in the paper and the responds to your comments are as flowing:
Points1: The authors have NOT clarified how urine was collected.
The answer is vague and does not explain the procedure followed. There are various methods for urine collection from sheep, so the method employed must be described clearly.
Response: We have added the details in the manuscript (line: 111-116):
At the end of the measurement period, urine sample was collected for lipid metabolism analysis: urine samples were collected on the morning of 60rd day of the feeding experiment, the collection device was made according to Kowalczyk et al. (1996) description. Transferred the urine to 50 ml sterile cryopreservation tube immediately after urination. Then urine samples were stored at −80 °C for later analysis.

Reviewer 2 Report
The paper has been improved.
Before acceptance, please include a new sub-section in the Discussion with the clinical significance of the findings.
Author Response
Response to Reviewer 2:
Dear Reviewer:
Thank you for your review and suggestions on our manuscript. The main corrections in the paper and the responds to your comments are as flowing:
Points 1: Before acceptance, please include a new sub-section in the Discussion with the clinical significance of the findings.
Response: We have added new sub-section in the discussion with the clinical significance of the study (445-465). The details are as follows:
4.6. The clinical significance of the study
Maintaining the balanced lipid metabolism is essential for both humans and animals. Lipids, as structural elements of cell membranes, could influence metabolic homeostasis through a variety of mechanisms and are necessary for the maintenance of metabolic homeostasis. As a necessary category of lipids, sphingolipids play vital roles in organisms. Complex sphingolipids can be converted into biologically energetic metabolites such as sphingosine, sphingomyelin and ceramide that play essential roles in promoting cell recognition, inhibiting proliferation, and stimulating apoptosis, signal transduction and cytokinesis [62,63]. Besides, the disordered lipid metabolism is also one of the essential elements influencing metabolic diseases [64,65]. In this study, the sheep in treatment group III showed some inflammatory reactions resulting from dysregulated lipid metabolism, which could be harmful to the health of the animal. Thus, the prevention of lipid metabolism disorders is necessary for both humans and animals.
Fatty acids play central roles in growth and development through their roles in membrane lipids, as ligands for receptors and transcription factors that regulate gene expression, precursor for eicosanoids, in cellular communication, and through direct interactions with proteins [66]. The addition of inulin and Chinese gallotannin in diet would change the content of fatty acids in sheep. A lower the content of SFA and a higher the content of UFA is good for sheep health, which is in line with the demand of producers and consumers.
